# The impact of COVID-19 on allied health professions

Jennifer Coto[1☯], Alicia Restrepo[2☯], Ivette Cejas[2☯‡], Sandra Prentiss[2☯‡*]

**1** Department of Pediatrics, University of Miami Miller School of Medicine, Miami, FL, United States of America, **2** Department of Otolaryngology Ear Institute, University of Miami Miller School of Medicine, Miami, FL, United States of America

☯ These authors contributed equally to this work.
‡ These authors are joint senior authors on this work.
* s.prentiss@med.miami.edu

**Data Availability Statement:** All relevant data are within the manuscript and its Supporting Information files.

**Funding:** The authors received no specific funding for this work.

## Abstract

The purpose of the current study was to examine the impact of Severe Acute Respiratory Syndrome Coronavirus 2 (SARS-CoV-2 or COVID-19) on allied health professionals work environment, access to personal protective equipment (PPE) and COVID-19 testing, and mental health. A 34-question survey was developed and distributed electronically to allied health professionals through listservs of professional organizations and social media groups. A total of 921 responses from allied health professionals in a variety of work settings were analyzed. The majority of allied health professionals had access to medical-grade PPE and agreed with their clinics decisions to stay open or closed. Private practices appeared to be the most negatively impacted with regards to employment in the form of pay reductions, furloughs, lay-offs, or the requirement of using paid time off. Importantly, 86% of all respondents, irrespective of employment status, reported feeling stressed with regards to changes in their work environment and transmission of the virus. However, levels of stress were dependent upon access to PPE and mental health resources. Specifically, those with access to mental health support reported lower stress levels than those without such access. These results highlight the need for continuous monitoring of mental health for allied health professionals in order to inform clinic and hospital policies for PPE and the development of brief interventions to mitigate adverse long-term mental health outcomes.

## Introduction

The World Health Organization (WHO) designated Severe Acute Respiratory Syndrome Coronavirus 2 (SARS-CoV-2 or COVID-19) as a global pandemic on March 11, 2020 [1]. The Centers for Disease Control and Prevention (CDC) reported that as of May 16, 2020 there were 1,435,098 total cases and 87,315 deaths as a result of COVID-19 in the United States [2]. For comparison, the CDC estimated 34,200 deaths due to influenza during the 2018–2019 season [3], suggesting that COVID-19 has a higher mortality rate than the seasonal flu in the previous year. Due to the rapid spread and contagious nature of this virus, the government mandated formal social distancing procedures, including school and business closures.

**Competing interests:** The authors have declared that no competing interests exist.

In April 2020, the Centers for Medicare and Medicaid Services (CMS) recommended limiting all elective, non-essential medical services to help reduce the spread of COVID-19 and to preserve personal protective equipment (PPE) for healthcare providers who were working directly with COVID-19 positive patients [4]. As a result, current unemployment rates have significantly increased in the healthcare industry [5]. Barnett, Mehrotra, and Landon [6] reported that due to COVID-19, hospitals have laid off employees and reduced salaries, ambulatory practices have considered closing down, and there is a possibility of more closures and bankruptcies. Similarly, orthopedic practices have been forced to limit staff hours, close offices, and reduce salaries during the COVID-19 pandemic greatly affecting allied health professionals within these practices [7].

In fact, 35–45% of respondents participating in survey studies have indicated that their medical practices were temporarily closed during COVID-19 [8, 9]. Those practices who remained open, were forced to reduce expenses by decreasing staff hours, furloughing or laying off employees, or implementing executive-level salary reductions [9]. Although the Storm et al. studies [8, 9] identified an important impact of COVID-19 on healthcare workers, they primarily surveyed private practice providers and only surveyed hearing care professionals (e.g., audiologists and hearing instrument specialists). To date, no study has compared the impact of COVID-19 across allied health professionals, as well as examined the impact of employment status on levels of stress. A better understanding is needed in terms of how employment status, access to PPE, and mental health supports are interrelated.

Evidence suggests that availability of PPE is associated with healthcare workers' willingness to work during an influenza public health emergency [10]. Despite the CDC and WHO guidelines regarding recommended PPE for healthcare workers, the rapid progression of COVID-19 resulted in a shortage and lack of PPE for healthcare workers [11]. This led to inappropriate reuse of disposable PPE and the use of non-medical-grade or homemade PPE [11, 12], which prior research has demonstrated is not adequate protection compared to medical-grade PPE [13, 14]. For many institutions, the decision to remain open for elective procedures or non-essential outpatient visits was coupled with the need to save PPE, including facemasks, goggles or face shields, and gloves [15]. One study surveying hearing health providers found that the most common safety measures that hearing practices were using to help reduce disease transmission during the COVID-19 pandemic included wiping down surfaces, using masks and gloves, and limiting the number of patients seen daily [8, 9]. However, these surveys did not inquire about the availability of PPE, nor examined the potential differences among allied health professionals or practice settings.

The literature on the impact of COVID-19 is steadily increasing with recent studies highlighting the need to respond to the psychological challenges during COVID-19 and calls for appropriate psychosocial support for healthcare workers. Healthcare workers are inherently subjected to increased stress due to their occupational environment [16], with a higher risk for emotional disturbances during infectious outbreaks [17, 18]. Previous research conducted during the Middle East Respiratory Syndrome coronavirus (MERS-CoV) outbreak found that 54.5% of healthcare workers surveyed had Post-Traumatic Stress Disorder (PTSD) symptoms, with 40% meeting criteria for diagnosis [19]. Additionally, during the Severe Acute Respiratory Syndrome (SARS) outbreak, healthcare workers on the frontlines reported fear and emotional distress [20, 21]. More importantly, these emotional symptoms persist for some individuals even after the outbreak has ended [19], with one study demonstrating that 18–57% of healthcare workers experienced emotional distress both during and after the outbreak [22]. These elevated symptoms are most commonly associated with a fear of contagion, concern for family health, and job stress [18, 20, 21, 23].

Thus, it is imperative that we begin screening healthcare works for increased levels of stress to identify those who would most likely benefit from intervention. This would help prevent the

occurrence of long-term psychopathology, including depression and PTSD. This is supported by previous research that recommends increased screening and surveillance of psychological symptoms in healthcare workers given the repeated exposures to traumatic experiences during a global health crisis [24].

The purpose of the current study is to expand on prior literature related to the impact of COVID-19 on healthcare workers, particularly with a focus on allied health professionals. This is the first study to examine the impact of COVID-19 on allied health professionals work environment, access to PPE, and levels of stress. Examining the interrelationships among these factors is crucial as we begin to transition to new models of service delivery, including telehealth or hybrid models of in-person and virtual clinics. By identifying the impact of the COVID-19 pandemic on allied healthcare professionals, we can inform policy and develop interventions to mitigate adverse long-term mental health outcomes.

## Methods

### Participants

Data was obtained via an online survey distributed electronically to allied health professionals through listservs of professional organizations and social media groups (see survey details below). The following exclusion criteria was used: 1) Individuals who do not read English, 2) Individuals working internationally, and 3) Individuals identifying as a non-allied health professional or identifying as an emergency medical technician (EMT). EMTs were excluded despite being considered allied health professionals due to the higher risk of infection and stress they encounter daily [25, 26]. EMTs typically function in a less controlled environment than other allied health professionals even outside of a pandemic [27]. Therefore, their responses would likely differ from other allied health professionals. A total of 1,171 responses were received over a one-week period and two-hundred and fifty responses were excluded due to not meeting inclusion criteria or partial survey completion. Participants were primarily female (88.3%) and 25–34 years of age (46.4%). Responses were obtained from allied health professionals working in 48 states (at least one response was obtained from every state except for Hawaii and Wyoming) and the District of Columbia (Fig 1).

Allied health professionals participating in this study included: Audiologists and Audiologist assistants (27.5%), Social Support Services (e.g, psychologists, mental health counselors, social workers; 27%), Other (e.g, nutritionists, dieticians, dental hygienists; 12.6%), Speech-Language Pathologists (SLPs) and SLP assistants (12.5%), Physical Therapists and Physical Therapist assistants (11.0%), and Occupational Therapists and Occupational Therapist assistants (9.4%). Over 50% of these professionals had worked in their current position for 0–5 years (56.6%) and the majority of respondents worked in a private practice (23.5%), university-hospital setting (20.3%), or hospital (19.5%). See Table 1 for further descriptive information.

### Procedures

This study was approved by the Institutional Review Board at the University of Miami. The survey was distributed via social media and electronic mail by the study co-investigators to relevant professional organizations who then distributed the survey to their members via listserv. The email sent contained a Qualtrics link where there was an informational letter explaining the study on the first page. The survey was open for seven days (4/16/2020 through 4/23/2020). If participants consented and chose to participate, they chose "next" and were given access to the survey sections. Participants were allowed to skip any question that they did not wish to answer. The consent specified that only allied health professionals should complete the survey.

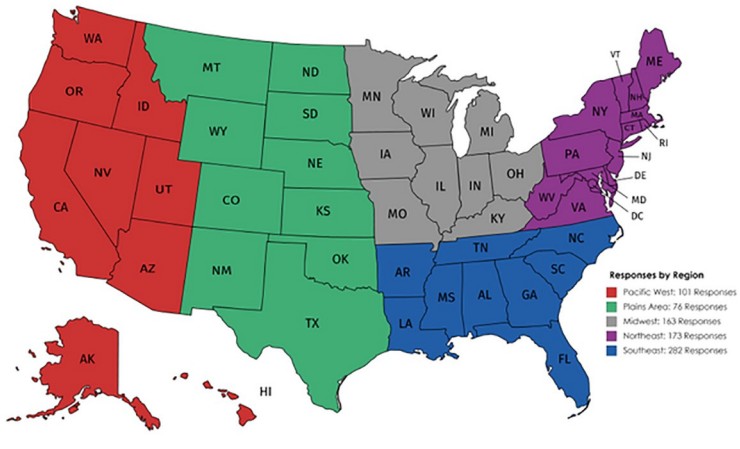

**Fig 1. Responses by region.** This figure illustrates the number of respondents by region. Reprinted from https://mapchart.net/ under a CC BY license, with permission from mapchart.net original copyright (2013).

The research team was comprised of audiologists, researchers, and psychologists experienced in social/behavioral research and instrument development. The survey was developed based on literature published to date [6–9, 17, 18], themes observed on social media groups for allied health professionals, and clinical experience. Additionally, the survey adhered to The Checklist for Reporting Results of Internet E-Surveys [CHERRIES; 27], including the use of adaptive questioning and handling of incomplete questionnaires. A 34-question survey was developed that focused on the impact of COVID-19 on three areas: work environment, access to personal protective equipment (PPE) and COVID-19 testing, and mental health (S1 File). We created this pilot study survey *de novo* as there are currently no established or validated measures of the impact of COVID-19 on allied health professionals.

In the work environment section of the survey, we asked participants questions related to their profession, years worked in their current position, and clinical setting. Additionally, questions related to the current status of their setting (e.g., if they are open, open in a limited capacity, closed) and changes in work responsibilities (e.g., practicing as usual, reassigned to perform tasks outside of normal clinical duties) were asked. Furthermore, questions related to the type of screening procedures and/or restrictions implemented by clinics were included. In the access to PPE and COVID-19 testing section, we asked whether participants had access to differing types of PPE in their clinical settings. We also inquired whether participants had been tested for COVID-19 and corresponding results. The mental health section queried whether participants had access to mental health support, which supports they had utilized, and the perceived importance of access to mental health support during the current pandemic. Additionally, concerns regarding acquisition and transmission of COVID-19 were also assessed.

## Data analysis

Analyses were conducted using the Statistical Package for Social Sciences, version 25 (SPSS, 2017). All complete responses were analyzed ($n$ = 921). Descriptive analyses were conducted for demographic variables and all survey items. Regression analyses between age, concern for

**Table 1. Respondent demographics.**

| Characteristic | N (%) |
|---|---|
| *Sex*: | |
| Female | 812 (88.3%) |
| Male | 104 (11.3%) |
| Other or Prefer not to answer | 4 (0.4%) |
| *Age*: | |
| 18–24 years | 23 (2.5%) |
| 25–34 years | 426 (46.4%) |
| 35–44 years | 223 (24.3%) |
| 45–54 years | 135 (14.7%) |
| 55–64 years | 80 (8.7%) |
| 65 years or older | 32 (3.5%) |
| *Household Size (Including Respondent)*: | |
| One or two | 514 (56%) |
| Three or Four | 339 (36.9%) |
| Five or Six | 62 (6.8%) |
| Seven or more | 3 (.3%) |
| *Profession*: | |
| Audiology | 253 (27.5%) |
| Social Support Services | 249 (27%) |
| Speech Therapy | 115 (12.5%) |
| Physical Therapy | 101 (11%) |
| Occupational Therapy | 87 (9.4%) |
| Other | 116 (12.6%) |
| *Position*: | |
| Staff | 550 (63.4%) |
| Faculty | 184 (21.2%) |
| Graduate Student or Student | 91 (10.5%) |
| Post-Doctoral Fellow | 43 (5%) |
| *Time working in current position*: | |
| 0–5 years | 521 (56.6%) |
| 6–10 years | 142 (15.4%) |
| 11–15 years | 95 (10.3%) |
| 16–20 years | 64 (6.9%) |
| 21 years or more | 99 (10.7%) |
| *Primary clinical setting* | |
| Private Practice | 216 (23.5%) |
| University- Hospital | 187 (20.3%) |
| University- Non-Hospital | 57 (6.2%) |
| Hospital | 180 (19.6%) |
| School/Educational Setting | 56 (6.1%) |
| Rehabilitation Center | 49 (5.3%) |
| Nursing Home | 40 (4.3%) |
| VA Hospital | 19 (2.1%) |
| Retail Setting | 7 (.8%) |
| Manufacturer | 5 (.5%) |
| Other | 104 (11.3%) |

transmission or acquisition of COVID-19, stress due to change in work environment, and use and access of mental health support, respectively, were conducted. Additionally, given the focus on the various allied health professions, we conducted multiple one-way ANOVA's to examine differences between professions on stress, importance of access to mental health support, and perception of being an essential worker.

## Results

### Work environment and access to PPE

The majority of respondents (67.2%) reported that they are currently employed, while 11.8% of respondents reported they are employed with a reduction in pay; 10.5% of participants responded that they have been furloughed and 4.2% of participants responded that they have been laid-off. A small percentage of respondents reported that they were required to take paid time off (PTO) with unpaid leave (3.9%) or allowed to use unaccrued PTO (2.4%). Audiologists (119 respondents) and physical therapists (49 respondents) reported being most affected in terms of their employment status, as well as professionals working in private practice.

At the time of the survey, 45.9% of respondents reported that their office or facility was open in a limited capacity (e.g., emergencies only) and 35.2% of respondents reported that their clinic was open. Of the survey participants who reported that their office is open or open in a limited capacity, 53.2% reported that they are seeing patients via telehealth, 42.9% reported that they are working from home, 37.8% reported that they are seeing patients on-site daily, and 31.2% reported that they are seeing emergent cases only. Of the respondents who reported that their office or facility is closed, 15.1% of respondents reported that their clinic is closed with plans to re-open and 3.8% reported that their clinic is closed with no plans to re-open. The majority of these respondents (75.3%) reported their office being closed for four or more weeks. Overall, providers reported agreeing with their clinics decision to remain open (90.1%) or closed (98.3%).

Interestingly, providers working from home reported a range of work responsibilities, including administrative activities (86.9%), updating clinical protocols (51.4%), research (41.3%), and publications (31.6%). Routine check-in with a supervisor was the most common measurement of productivity reported (62.2%), followed by time tracking (35%), and project management trackers (17.5%).

Ninety percent of allied health professionals continued to see patients under their scope of practice, while 10% were reassigned to perform other duties. Tasks that respondents were reassigned to included screening patients (2.1%), taking temperatures (2.1%), scheduling (1.6%), or triaging patients (1.1%). Respondents also indicated "other" responses, such as, attending trainings, academic work, and working in the COVID-19 call center to screen employees.

Our data also supported that essentially all clinics and hospitals were conducting some form of screening for COVID-19. The most common responses included implementing verbal screenings (e.g., asking patients "Do you have a cough?") and physical screenings (e.g., taking temperatures) with response rates of 74.9% and 57.6%, respectively. Physical distancing (e.g., drop-box for devices; 38.5%), and limiting patients to one companion (30.8%) or no companions (30.7%) were also frequently endorsed. Other common responses included mandatory masks for staff and visitors and limiting the amount of time spent engaging with patients.

The majority of respondents reported having access to PPE at work with 12.8% of healthcare professionals reporting no access to PPE at work. Those with access reported availability of surgical masks (60.5%), N-95 masks (27.6%), homemade facemasks (25.7%), and surgical masks with face shields (19.8%). Mental health specialists (27 respondents) and audiologists (17 respondents) were the most common respondents reporting no access to PPE at work. Nearly half (49.2%) of respondents reported no access to COVID-19 testing at work, while

33% reported availability of testing, and 17% were unsure. At the time of the survey, over 95% of respondents had not been tested for COVID-19 or COVID-19 antibodies; however, of those who were tested, 23.3% tested positive and (14.4%) were waiting on results.

## Mental health

Allied health professionals were also queried regarding their stress related to changes in their clinical practice during the pandemic. The majority of respondents reported that they either agreed (48.7%) or strongly agreed (37.5%) with the statement that they felt stressed due to the changes in clinical activity. Participants were also asked to rank their level of concern about acquiring COVID-19 at work and outside of work (e.g., grocery stores, pharmacies, outdoor exercise) as well as their concern regarding transmitting the virus to their family members or other individuals. Although providers were concerned about acquiring COVID-19 at work (75.6%) a larger percentage of our respondents reported being concerned about acquiring COVID-19 outside of work (97.4%). Additionally, the majority of respondents (96.2%) reported concern regarding transmission of COVID-19 to others (Fig 2).

Age was a significant factor in respondents concern about transmitting COVID-19 to family or others, $F(1,916) = 10.25$, $p < .001$, $R^2 = .01$, $\beta = .11$, with older individuals reporting more concern about transmitting COVID-19 to their family or others. No significant associations were found between age and concern about acquiring COVID-19 at work, or in other places, respectively.

Interestingly, despite allied health professionals reporting significant levels of stress during the pandemic, few providers were utilizing any form of mental health support. The majority of allied health professionals (93%) reported access to mental health services was very important or important during this time, with approximately 75% having access to a method of mental health support. The most common mental health supports included mental health phone applications (53.5%), followed by mental health webinars (40.8%), and mental health counselor/psychologists (39.5%). Despite the importance placed on access to mental health support, few respondents reported utilizing any method of mental health support (Fig 3).

In terms of stress regarding the change in clinical practice due to COVID-19, there was a significant association between stress and access to mental health support, with those who had more access feeling less stressed, $F(1,916) = 17.07$, $p < .001$, $R^2 = .02$, $\beta = .14$. However, there was no significant association between level of stress and *use* of mental health services, indicating that

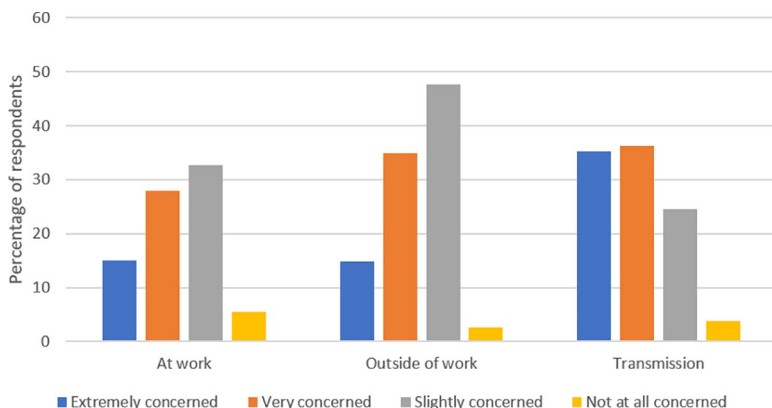

**Fig 2. Concern regarding acquisition and transmission of COVID-19.** This figure illustrates allied health professionals' level of concern of acquiring COVID-19 at work or outside of work, as well as transmission to others. A larger percentage of respondents reported high levels of concern regarding transmission of COVID-19 to others.

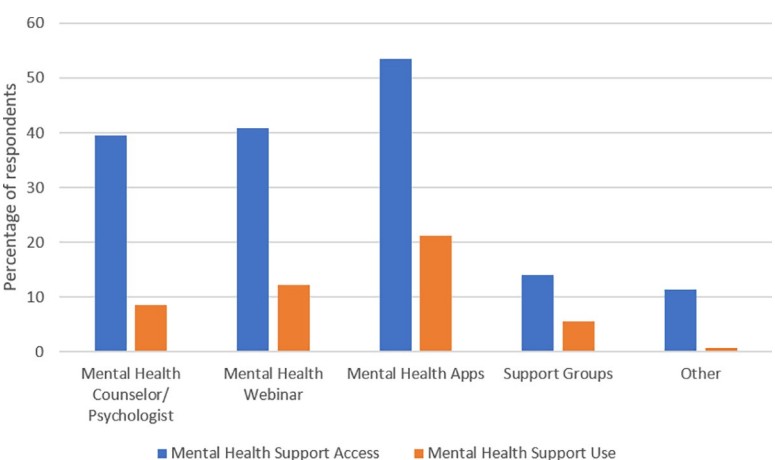

**Fig 3. Access and use of mental health support.** This figure represents the discrepancy in access and use of mental health supports. Phone applications were the most accessible and utilized method of mental health support.

regardless of use of mental health services, an individuals' reported level of stress remained the same. Additionally, access to PPE was significantly associated with level of stress, with those reporting no access to PPE having higher levels of stress $F(1,712) = 3.94$, $p < .05$, $R^2 = .01$, $\beta = -.07$.

### Differences between professions

Given the variability in clinical responsibilities for the allied health professionals included in this study, we examined differences in stress levels by profession. There was a statistically significant difference between groups as determined by one-way ANOVA of stress by profession $(F(5,912) = 4.51$, $p < .001)$. Bonferroni post-hoc tests revealed that providers working in social support services ($M = 1.95$, $SD = .72$) reported feeling less stressed than audiology ($M = 1.69$, $SD = .70$) and speech ($M = 1.67$, $SD = .65$) providers.

Additionally, there was a statistically significant difference between professionals on the importance of access to mental health services ($F(5,915) = 10.12$, $p < .001$). Bonferroni post-hoc tests revealed that the social support services respondents ($M = 1.30$, $SD = .52$) were significantly different than all other professions, with social support providers endorsing higher ratings of agreement that mental health access is important during this time [physical therapists ($M = 1.71$, $SD = .68$), OT ($M = 1.60$, $SD = .66$), audiology ($M = 1.60$, $SD = .59$), other ($M = 1.58$, $SD = .70$), and speech ($M = 1.57$, $SD = .62$)].

Lastly, there was a statistically significant difference between professionals on whether they perceived their profession was essential ($F(5,914) = 36.99$, $p < .001$). Bonferroni post-hoc tests revealed that audiologists ($M = 2.13$, $SD = .49$) were significantly different than all surveyed professionals, with audiologists endorsing lower ratings of agreement than all other professions [other ($M = 1.76$, $SD = .74$), speech ($M = 1.63$, $SD = .54$), PT ($M = 1.59$, $SD = .51$), OT ($M = 1.55$, $SD = .55$), and social support services ($M = 1.53$, $SD = .53$)]. Additionally, there was a significant difference between the social support providers ($M = 1.53$, $SD = .53$) and "other" professions ($M = 1.76$, $SD = .74$), with social support respondents endorsing higher ratings of agreement that their profession is essential.

### Discussion

At the end of 2019, a novel coronavirus was identified as the cause of a cluster of pneumonia cases in Wuhan, which rapidly spread, resulting in a worldwide pandemic [1]. Healthcare

workers across the country quickly responded to the public health emergency caused by SARS-CoV-2 or COVID-19. The CDC and WHO provided guidance for frontline healthcare workers regarding personal protective equipment, primarily focusing on healthcare providers who have direct contact with COVID-19 positive patients [11, 28, 29]. Little guidance and attention were initially given to healthcare professionals working alongside frontline workers, including allied health professionals (i.e., audiologists, speech and language therapists, psychologists, physical therapists, occupational therapists, etc.) and PPE was initially being saved for those working with COVID-19 positive patients [29]. This is the first study to examine the impact of COVID-19 on allied health professionals work environment, access to PPE, and levels of stress.

Overall, the majority of allied health professionals were still employed during the pandemic with 89% of practices open in some capacity. However, 14.7% of participants had been furloughed or laid-off, likely to mitigate the financial crisis that many institutions and private practices rapidly faced. Providers also generally agreed with their clinics decision to remain open or closed and the majority reported having access to PPE, which affected professionals stress related to working.

During this pandemic, healthcare delivery models were quickly challenged and many university and private practices implemented telehealth or virtual clinics to accommodate patient needs [30], with the uncertainty of whether reimbursement would be obtained for telehealth services [31]. Emergency legislation was passed which allowed for flexibility in reimbursement for telehealth with some clinics quickly adopting this new model of service delivery. Specifically, CMS provided waivers that allowed practitioners to be reimbursed for services provided via telehealth during the pandemic [32]. Our data supports the rapid adoption of telehealth, with 43% of providers engaging in telehealth services. Despite the initial challenges experienced by many institutions and providers on implementing telehealth, including training for providers and obtaining HIPAA compliant programs, telehealth has proven to be a successful method for obtaining case histories, screening patients in which it may be unclear if they need to be seen, and providing counseling services. Moreover, telehealth may serve as a possible solution to healthcare systems that are already stressed and looking for ways to increase clinical volume and reduce costs (e.g., overhead, supplies). As suggested by Smith and colleagues [33], continued monitoring of telehealth policies and funding will be necessary as insurances and policy makers determine whether temporary reimbursement of telehealth services will remain indefinitely following the pandemic.

Further, as hospital and institution leaders develop new models for clinical care, including hybrid models where some patients are seen in-person while others continue to be served via telehealth, it will be increasingly important to understand and assess how providers are feeling related to their new schedules and work environments. Notably, our data found that allied health professionals reported concerns about acquiring COVID-19 both at work and outside of work, with the greatest concern about transmitting the virus to others, especially in older respondents. It may be the case that older respondents have more interactions with vulnerable populations, such as taking care of elderly parents. These results are consistent with prior studies, which demonstrated healthcare worker concerns regarding acquiring disease and disease transmission to family [19–21, 23]. Institutions and decision-makers need to be aware of these concerns and provide alternate work environments if available, including telecommuting options or hybrid models allowing for both staggered in-person and virtual clinics.

Some of the concerns about transmission may be ameliorated by access and use of PPE, as prior research has found that access to PPE is associated with willingness to work [10]. The majority of allied health professionals surveyed reported having access to PPE (87.2%), with 7% only having access to non-medical grade PPE (e.g., homemade masks). This is

contradictory to previous reports of PPE shortage [34] and highlights recent government and hospital efforts to provide access to PPE for all individuals interacting with patients.

This is also consistent with the WHO's Recommendations for the Rational Use of Personal Protective Equipment, which provides guidance on appropriate PPE use according to the setting, personnel, and type of activities [29]. Interestingly, there is no direct guidance for allied health professionals. The closest appropriate guidance is the outpatient facilities triage for healthcare workers conducting preliminary screening not involving direct contact category which recommends maintaining a spatial distance of at least 1 meter and no required PPE [29]. Some may interpret these WHO guidelines as suggesting that PPE is not required for allied health professionals [29]; however, our data supports that institutions and clinical practices may be implementing stricter guidelines on the use of PPE for all providers. Note, although our findings suggest that the majority of allied health professionals had *access* to PPE, we did not inquire if they were using the PPE and under what circumstances.

In addition, our findings linked access to PPE with reported levels of stress, with allied health professionals reporting no access to PPE endorsing higher levels of stress. This suggests that although it may not be required for some allied health professionals to use PPE, access to PPE helps to mitigate provider stress levels and willingness to come to work. This is consistent with prior literature reporting that healthcare workers are more likely to work during a pandemic if they feel confident that the hospital can protect them [35].

Moreover, given the substantial reports of provider stress during the COVID-19 pandemic in this study, it is important that institutions implement screening for symptoms of mental health disorders as early identification of heightened levels of stress may serve to prevent future long-term psychopathology. Healthcare workers are naturally subjected to increased stress secondary to their occupational environments [16]; however, increased feelings of fear and emotional distress as well as symptoms of PTSD have been reported during previous health crises, such as the MERS-CoV and SARS outbreaks [20–22]. Our current study identified that 86.2% of allied health professionals were reporting stress related to their employment.

Understanding of provider current levels of stress is crucial in order to better prepare for the emotional sequela that takes place during and after a pandemic. Previous research has suggested that there are four waves during a pandemic which include an increase in anxiety symptoms, then depressive symptoms, followed by more long-term psychological effects, such as depressive disorders and PTSD [36]. Our findings suggest an increase in stress for all the professions surveyed, with the social support services group feeling less stressed than both the audiology and speech and language pathology groups. It is possible that those in the social support services group have better developed peer networks that include mental health specialists. In addition, given their clinical training and knowledge, these mental health providers likely have better self-awareness to identify their symptoms and when further support is needed.

Early identification and intervention for mental health symptoms is crucial, particularly during health crises in which we have the opportunity to intervene and prevent long-term psychopathology, such as depressive disorders and PTSD. Given the current findings of increased stress due to COVID-19, it is imperative to screen for these symptoms both during and after a pandemic. This is necessary in order to identify and support those with elevated symptoms as early and swiftly as possible. Screening is particularly important as access to typical resources and supports are limited during pandemics. This is supported by prior research that recommends increased screening and surveillance of psychological symptoms in healthcare workers given the repeated exposures to traumatic experiences [24]. These screenings will also support development of brief evidenced-based interventions.

As suggested by Onyeaka and colleagues [37] little is known about how to respond to the psychological challenges already being experienced during COVID-19. The authors highlight the

need for appropriate psychosocial support. Our study demonstrated that the majority of respondents had access to mental health supports and those who had more access reported feeling less stress. However, there were no associations between level of stress and use of such supports. This indicates that knowing resources are available may be enough to reduce the general stress associated with the clinical changes due to COVID-19. Institutions and organizations should continue to offer these support services to their faculty and staff so that providers can be aware of the services available. This would also assist with normalizing the need for mental health support during a highly stressful time. With most professionals reporting access and use of mental health phone applications or webinars, it is recommended that prioritization be given to the development of brief interventions that may be conducted virtually around providers schedules.

## Limitations & future directions

Although this study was the first to assess the impact of the pandemic on allied healthcare providers' work environment and stress, it also had some limitations that are important to consider. First, although we did inquire whether respondents had *access* to PPE, we did not inquire if they were *using* the PPE and under what circumstances. Future research would benefit from gathering information on PPE use as this can be helpful for inventory planning and policy making. Despite this, the current study contributes to the literature as it identified that having access to PPE helped mitigate reported levels of stress. Second, this study was conducted during the beginning of the pandemic and therefore results may be different if the survey were administered following a prolonged period of time. However, identifying how providers are feeling at the beginning of a pandemic is important as it can inform decisions regarding need for monitoring or interventions, as well as policies for allied healthcare providers working in different settings. In addition, our study did not include all allied health professionals. For example, EMTs were excluded due to their higher risk of exposure and stress. Future research should examine differences among all allied health professionals, including those who are at higher risk. Lastly, our sample consisted of primarily audiology and social support providers and therefore the results may not generalize to the entire population of allied health professionals. Yet, we did have a breadth in our sample in terms of geographic representation (respondents from 48 states) and work settings (e.g., private, university-based, hospital). We also received responses from professionals from various disciplines allowing us to compare responses by profession.

## Conclusion

In conclusion, our study is among the first to examine how COVID-19 is impacting work environments and provider stress. Our data highlighted the rapid adoption of telehealth and virtual clinics by allied health professionals, illustrating that hybrid models of service delivery are feasible for the future. It was also evident that healthcare providers are willing to provide clinical care even under stressful environments. However, access to PPE and mental health supports were related to providers' levels of stress. Thus, PPE should continue to be offered to allied health professionals providing in-person clinical care. More importantly, the identified heightened levels of stress reported by professionals warrants further attention. Based on our results, continued monitoring of stress and mental health screening is strongly recommended. Early identification of these mental health symptoms will support provider well-being, decrease burn-out, and prevent long-term psychopathology.

## Supporting information

**S1 File. Survey.** This file includes the survey used in the current study.
(PDF)

## Author Contributions

**Conceptualization:** Alicia Restrepo, Ivette Cejas, Sandra Prentiss.

**Data curation:** Jennifer Coto, Ivette Cejas.

**Formal analysis:** Jennifer Coto, Ivette Cejas.

**Investigation:** Jennifer Coto, Ivette Cejas, Sandra Prentiss.

**Methodology:** Jennifer Coto, Alicia Restrepo, Ivette Cejas, Sandra Prentiss.

**Project administration:** Alicia Restrepo, Ivette Cejas, Sandra Prentiss.

**Supervision:** Ivette Cejas, Sandra Prentiss.

**Validation:** Ivette Cejas, Sandra Prentiss.

**Visualization:** Jennifer Coto, Ivette Cejas, Sandra Prentiss.

**Writing – original draft:** Jennifer Coto, Alicia Restrepo, Ivette Cejas, Sandra Prentiss.

**Writing – review & editing:** Jennifer Coto, Ivette Cejas, Sandra Prentiss.

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
