## [Decision Letter · Decision Letter 0]

14 Jul 2020

PONE-D-20-15411

The Impact of COVID-19 on Allied Health Professions

PLOS ONE

Dear Dr. Prentiss,

Thank you for submitting your manuscript to PLOS ONE. After careful consideration, we feel that it has merit but does not fully meet PLOS ONE’s publication criteria as it currently stands. Therefore, we invite you to submit a revised version of the manuscript that addresses the points raised during the review process.

Please address the methodological issues raised by both reviewers.  Please also strengthen your discussion section supported by citations to empirical research.

We look forward to receiving your revised manuscript.

Kind regards,

Rosemary Frey

Academic Editor

PLOS ONE

Journal Requirements:

2. Please include additional information regarding the survey or questionnaire used in the study and ensure that you have provided sufficient details that others could replicate the analyses.

For instance, if you developed a questionnaire as part of this study and it is not under a copyright more restrictive than CC-BY, please include a copy, in both the original language and English, as Supporting Information. Moreover, please include more details on how the questionnaire was pre-tested, and whether it was validated.

3. In your Methods section, please provide additional information about the participant recruitment method and the demographic details of your participants.

Please ensure you have provided sufficient details to replicate the analyses such as:

a) the recruitment date range (month and year),

b) a description of any inclusion/exclusion criteria that were applied to participant recruitment,

c) a table of relevant demographic details,

d) a statement as to whether your sample can be considered representative of a larger population,

e) a description of how participants were recruited.

4.We note that Figure 1 in your submission contains map images which may be copyrighted.

We require you to either (a) present written permission from the copyright holder to publish these figure specifically under the CC BY 4.0 license, or (b) remove the figure from your submission:

b. If you are unable to obtain permission from the original copyright holder to publish these figure under the CC BY 4.0 license or if the copyright holder’s requirements are incompatible with the CC BY 4.0 license, please either i) remove the figure or ii) supply a replacement figure that complies with the CC BY 4.0 license. Please check copyright information on all replacement figures and update the figure caption with source information. If applicable, please specify in the figure caption text when a figure is similar but not identical to the original image and is therefore for illustrative purposes only.

Reviewers' comments:

Reviewer's Responses to Questions

**Comments to the Author**

1. Is the manuscript technically sound, and do the data support the conclusions?

Reviewer #1: Yes

Reviewer #2: Partly

2. Has the statistical analysis been performed appropriately and rigorously? 

Reviewer #1: Yes

Reviewer #2: Yes

3. Have the authors made all data underlying the findings in their manuscript fully available?

Reviewer #1: Yes

Reviewer #2: No

4. Is the manuscript presented in an intelligible fashion and written in standard English?

Reviewer #1: Yes

Reviewer #2: Yes

5. Review Comments to the Author

Reviewer #1: Thank you for the opportunity to review this manuscript. The topic will be of great interest to clinical leaders and policy makers. Overall the article is very well written and the suggestions I have made are minor. I hope the authors find the feedback helpful.

There are a few typos throughout that need correcting e.g. abstract: page 2 line 37 'of' is missing from the sentence "...impacted with regards to employment in the form pay reductions...."

1. Please ensure a copy of the survey is available in supplementary information.

2. Methods - how many surveys were sent out in total? What was the response rate?

3. Please provide a more comprehensive rationale for excluding EMT's given that they are considered allied health professionals?

4. Given the variation in COVID19 cases. transmision and public health response by region, how might this have impacted on the regional response rate?

5. On page 8 line 157 the authors state that they developed the survey based on literature - what literature specifically? Please include references to the literature used. What clinical experience did the research team have and how did this inform the development of the survey?

6. Throughout the discussion there are large pieces of text that do not have references. For example page 15 lines 296-298 "The CDC and WHO provided guidance for frontline healthcare workers regarding PPE, primarily focusing on physicians and nurses". Please reference the guidelines.

The statement that PPE was initially being saved for those working with positive COVID-19 patients. Can this be referenced? How was this information found?

Page 15 lines 313-315 "Emergency legislation was passed which allowed for flexibility..." Please reference the legislation

Uncertainty regarding reimbursement was referred to - can this be referenced with the information source?

The discussion would benefit from references to government or local guidelines that support some of the arguments used.

Reviewer #2: This is an interesting paper but would be strengthened if the authors reported findings against the CHERRIES checklist (Equator network) https://www.ncbi.nlm.nih.gov/pmc/articles/PMC1550605/?report=reader#!po=50.0000

There is no real limitations section and some of the claims in conclusion and discussion need modifying

6. PLOS authors have the option to publish the peer review history of their article (what does this mean?). If published, this will include your full peer review and any attached files.

Reviewer #1: No

Reviewer #2: No

---

## [Author Response · Author response to Decision Letter 0]

29 Jul 2020

REVIEWER COMMENTS

(Editor) RESPONSE

1. Please also strengthen your discussion section supported by citations to empirical research 

Response: Thank you for your feedback. We have strengthened our discussion by including a more comprehensive literature search and citations. 

2. Please address the methodological issues raised by both reviewers

Response:We have addressed the methodological issues raised by the reviewers and discussed in our limitations section. 

3. Please include additional information regarding the survey or questionnaire used in the study and ensure that you have provided sufficient details that others could replicate the analyses.

For instance, if you developed a questionnaire as part of this study and it is not under a copyright more restrictive than CC-BY, please include a copy, in both the original language and English, as Supporting Information. Moreover, please include more details on how the questionnaire was pre-tested, and whether it was validated. 

Response: We have added more detail regarding our survey development and have also uploaded the survey as supporting information. The survey was developed by the research team using published research available at the time of this study. It was not validated prior to use in this survey study. Lines 166-169

Response: We have reviewed the manuscript to ensure that it is consistent with the PLOS ONE style requirements and guidelines. 

Response: Files have been saved according to PLOS ONE’s style requirements. 

5. In your Methods section, please provide additional information about the participant recruitment method and the demographic details of your participants.

Please ensure you have provided sufficient details to replicate the analyses such as:

a) the recruitment date range (month and year),

b) a description of any inclusion/exclusion criteria that were applied to participant recruitment,

c) a table of relevant demographic details,

d) a statement as to whether your sample can be considered representative of a larger population,

e) a description of how participants were recruited. 

RESPONSE: We have specified that the survey was open for seven days (4/16/2020 through 4/23/2020; see line 151). Exclusion criteria was also made clearer in the methods section. Demographic information of the survey participants is included in Table 1 and a limitations section has been added, which includes a generalizability statement. Lastly, we added more detailed information regarding participant recruitment. Lines 128-136; 161; 432-437

6. We note that Figure 1 in your submission contains map images which may be copyrighted 

RESPONSE: We have reviewed the copyright for the map utilized in the manuscript. According to https://mapchart.net/feedback.html the image may be use for private or commercial use/publications freely as long as mapchart.net is referenced in the work. We have referenced mapchart.net in Figure 1. Please see screenshot from the website at the bottom of this document. The figure has the copyright information on the bottom left. 

7. PLOS requires an ORCID iD for the corresponding author in Editorial Manager 

RESPONSE: The ORCID ID for Dr. Prentiss is 0000-0001-7195-7254 

REVIEWER COMMENTS

(Reviewer 1) RESPONSE

1. There are a few typos throughout that need correcting e.g. abstract: page 2 line 37 'of' is missing from the sentence "...impacted with regards to employment in the form pay reductions...."

RESPONSE: We have reviewed the entire manuscript and have corrected any typos and suggested revisions. 

2. Please ensure a copy of the survey is available in supplementary information. 

RESPONSE: We have included the survey as supplementary information. Included and submitted as S1_File.pdf

3. Methods - how many surveys were sent out in total? What was the response rate? 

RESPONSE: Surveys were distributed via professional organization email listservs and social media; therefore, the response rate is unable to be calculated. 

4. Please provide a more comprehensive rationale for excluding EMT's given that they are considered allied health professionals?

RESPONSE: We have added a more detailed justification, including literature on our decision to exclude EMTs. As noted in the literature, EMTs are at a higher risk of infection and stress even in the absence of a pandemic (Orellana et al., 2016; Bentley et al., 2013). Thus, there responses would have likely differed significantly than other allied healthcare professionals. We have added this as a limitation and also recommended that future research examine allied health professionals who are at higher risk, such as EMTs. Lines 130-134

5. Given the variation in COVID19 cases. transmission and public health response by region, how might this have impacted on the regional response rate? 

RESPONSE: We appreciate the reviewer’s thoughtful feedback. We understand the concern about the transmission rates by region; however, this survey study was completed April 16-23 during the same time that the Center for Medicare and Medicaid Services recommended limiting all non-essential medical services. In addition, although we were able to capture respondents across all regions in the US, we do not have a large enough sample size in each region to examine differences among them. 

6. On page 8 line 1 the authors state that they developed the survey based on literature - what literature specifically? Please include references to the literature used. What clinical experience did the research team have and how did this inform the development of the survey? 

RESPONSE: The research team is made up of clinical audiologists, psychologists, and researchers. Psychologists are trained in social/behavioral research, as well as instrument development. Dr. Cejas has specifically published on instrument development and is a NIH researcher, who has development measures. As requested, we have also identified the research that was reviewed and used to develop the survey used in this study. We have ensured that the citations are included (6-9, 16-17). Lines 166-169

7. Throughout the discussion there are large pieces of text that do not have references. For example page 15 lines 296-298 "The CDC and WHO provided guidance for frontline healthcare workers regarding PPE, primarily focusing on physicians and nurses". Please reference the guidelines.

The statement that PPE was initially being saved for those working with positive COVID-19 patients. Can this be referenced? How was this information found?

Page 15 lines 313-315 "Emergency legislation was passed which allowed for flexibility..." Please reference the legislation

Uncertainty regarding reimbursement was referred to - can this be referenced with the information source?

The discussion would benefit from references to government or local guidelines that support some of the arguments used. RESPONSE: We have thoroughly reviewed the discussion and added any necessary references. 

REVIEWER COMMENTS

(Reviewer 2) RESPONSE

1. This is an interesting paper but would be strengthened if the authors reported findings against the CHERRIES checklist (Equator network) https://www.ncbi.nlm.nih.gov/pmc/articles/PMC1550605/?report=reader#!po=50.0000 Thank you for your comment and the reference. 

RESPONSE: The survey indeed followed that of the CHERRIES checklist in design, distribution, time frame for data collection and statistical analysis. This is explained in the methods section. We could not calculate a response rate as the surveys were listed on websites and social media. Therefore, we do not have access to the number of people who had access to the survey and did not complete. 

2. There is no real limitations section and some of the claims in conclusion and discussion need modifying

RESPONSE: We have added a limitations section as requested. We have also reviewed our conclusions to ensure that they are all supported by the data. Lines 417-437;

---

## [Decision Letter · Decision Letter 1]

7 Sep 2020

PONE-D-20-15411R1

The Impact of COVID-19 on Allied Health Professions

PLOS ONE

Dear Dr. Coto,

Thank you for submitting your manuscript to PLOS ONE. After careful consideration, we feel that it has merit but does not fully meet PLOS ONE’s publication criteria as it currently stands. Therefore, we invite you to submit a revised version of the manuscript that addresses the points raised during the review process.

Please make the minor revisions requested by Reviewer 2.

We look forward to receiving your revised manuscript.

Kind regards,

Rosemary Frey

Academic Editor

PLOS ONE

Reviewers' comments:

Reviewer's Responses to Questions

**Comments to the Author**

1. If the authors have adequately addressed your comments raised in a previous round of review and you feel that this manuscript is now acceptable for publication, you may indicate that here to bypass the “Comments to the Author” section, enter your conflict of interest statement in the “Confidential to Editor” section, and submit your "Accept" recommendation.

Reviewer #2: (No Response)

Reviewer #3: All comments have been addressed

2. Is the manuscript technically sound, and do the data support the conclusions?

Reviewer #2: Yes

Reviewer #3: Yes

3. Has the statistical analysis been performed appropriately and rigorously? 

Reviewer #2: (No Response)

Reviewer #3: I Don't Know

4. Have the authors made all data underlying the findings in their manuscript fully available?

Reviewer #2: Yes

Reviewer #3: No

5. Is the manuscript presented in an intelligible fashion and written in standard English?

Reviewer #2: Yes

Reviewer #3: Yes

6. Review Comments to the Author

Reviewer #2: If you have complied with the CHERRIES checklist for surveys, you need to reference this and note this intext. I have not changed the revised manuscript so nothing to upload.

Reviewer #3: Thank you for the opportunity to review this article again. The authors have responded to the reviewer’s feedback very well. I have nothing more to add.

7. PLOS authors have the option to publish the peer review history of their article (what does this mean?). If published, this will include your full peer review and any attached files.

Reviewer #2: No

Reviewer #3: No

---

## [Author Response · Author response to Decision Letter 1]

9 Sep 2020

We included a statement regarding the use of the CHERRIES checklist within the text and as a reference.

---

## [Decision Letter · Decision Letter 2]

14 Oct 2020

The Impact of COVID-19 on Allied Health Professions

PONE-D-20-15411R2

Dear Dr. Prentiss,

We’re pleased to inform you that your manuscript has been judged scientifically suitable for publication and will be formally accepted for publication once it meets all outstanding technical requirements.

Kind regards,

Rosemary Frey

Academic Editor

PLOS ONE

Additional Editor Comments (optional):

Reviewers' comments:

Reviewer's Responses to Questions

**Comments to the Author**

1. If the authors have adequately addressed your comments raised in a previous round of review and you feel that this manuscript is now acceptable for publication, you may indicate that here to bypass the “Comments to the Author” section, enter your conflict of interest statement in the “Confidential to Editor” section, and submit your "Accept" recommendation.

Reviewer #2: All comments have been addressed

Reviewer #3: All comments have been addressed

2. Is the manuscript technically sound, and do the data support the conclusions?

Reviewer #2: Yes

Reviewer #3: Yes

3. Has the statistical analysis been performed appropriately and rigorously? 

Reviewer #2: Yes

Reviewer #3: I Don't Know

4. Have the authors made all data underlying the findings in their manuscript fully available?

Reviewer #2: Yes

Reviewer #3: Yes

5. Is the manuscript presented in an intelligible fashion and written in standard English?

Reviewer #2: Yes

Reviewer #3: Yes

6. Review Comments to the Author

Reviewer #2: Thanks for following up, it makes for a more rigorous report.

Reviewer #3: (No Response)

7. PLOS authors have the option to publish the peer review history of their article (what does this mean?). If published, this will include your full peer review and any attached files.

Reviewer #2: No

Reviewer #3: No

---

## [Editor Report · Acceptance letter]

20 Oct 2020

PONE-D-20-15411R2 

The impact of COVID-19 on allied health professions 

Dear Dr. Prentiss:

I'm pleased to inform you that your manuscript has been deemed suitable for publication in PLOS ONE. Congratulations! Your manuscript is now with our production department. 

Kind regards, 

on behalf of

Dr. Rosemary Frey 

Academic Editor

PLOS ONE